# Monte Carlo Computer Simulations of Spin-Transfer Torque

**DOI:** 10.3390/ma16206728

**Published:** 2023-10-17

**Authors:** Sergey V. Belim, Igor V. Bychkov

**Affiliations:** 1Department of Physics, Omsk State Technical University, 644050 Omsk, Russia; 2Department of Radiophysics and Electronics, Chelyabinsk State University, 454001 Chelyabinsk, Russia; bychkov@csu.ru

**Keywords:** spin-transfer torque, multilayer films, spin-polarized current, computer simulation

## Abstract

This article performs computer simulations of the change in magnetization in the ferromagnetic film when polarized electric current passes through it. The model examines multilayer structures from ferromagnetic and nonmagnetic films. A sandwich system comprises two ferromagnetic layers separated by a nonmagnetic gasket. Ferromagnetic films have different magnetic susceptibility. The first ferromagnetic film is magnetically hard and acts as a fixed layer. The second ferromagnetic film is magnetically soft, with a switched direction of magnetization. The current direction is perpendicular to the film plane (CPP geometry). Spin transfer is carried out by electrons that polarize in the first ferromagnetic film and transmit spin to the second ferromagnetic film. We use the Ising model to describe the magnetic properties of the system and the Metropolis algorithm to form the thermodynamic states of the spin system. Simulations are performed at temperatures below the Curie points for both materials. The result of computer simulation is the dependence of magnetization in the magnetically soft film on the current strength in the system. Calculations show that there is a critical value of the current at which the magnetization sign of the controlled film changes. The magnetization versus current plot is stepwise. The change in the magnetization sign is due to an increase in the polarization of the electron gas. The plot of electron gas polarization versus current is also stepwise.

## 1. Introduction

The spin transport effect is the effect of magnetization in one film on magnetization in another by electric current. The main mechanism of this phenomenon is the transfer of the spin by polarized electric current. Experiments demonstrate several phenomena caused by spin transfer by electric current: magnetization switching, stationary magnetization oscillations, and movement of domain walls [1]. Spin transport provides a means to control the magnetization in thin films and is an important tool for spintronic devices.

The passage of spin current in multilayer spin systems has been actively investigated over the past decades [2,3,4]. This interest is based on the use of the giant magnetic resistance effect to store data and on magnetic sensors [2,3]. Experimental studies show that electrons with spins parallel to the spins of the medium meet less resistance. A large number of parameters affect the spin current. Interface roughness leads to electron scattering and changes giant magnetoresistance [5,6]. Spin current dynamics also significantly depend on the system temperature [7]. Several factors affect the magnetoresistance of an electric current: scattering of stray spins on lattice magnons [8], diffusion on impurities [9], and spin–spin correlation [10]. Different materials show different dependence of magnetic resistance to electric current near Curie temperature. Some materials have a high peak on the resistance plot at the phase transition temperature [11]. Other materials do not have a peak on the resistance plot but show a peak on the temperature derivative plot [12].

In multilayer structures, the current perpendicular to layers (CPP geometry) is polarized by one ferromagnetic layer and transmits the spin moment to another ferromagnetic layer (spin-transfer torque) [13]. Large current densities are necessary to realize this effect (>10^7^ A/sm^2^). Spin-transfer torque is implemented in three-layer systems consisting of two ferromagnetic films separated by nonmagnetic metal. The gasket thickness is smaller than the spin diffusion length. Electrons do not change the direction of their spin in this case. The second layer is more easily magnetized than the first layer. This layer is called free. The first layer is magnetically hard. It is called fixed. Spin current equalizes the magnetization in the free layer in the direction of magnetization in the fixed layer. The reverse direction of the current leads to the antiparallel orientation of the magnetizations. Spin filtration occurs on the interface of ferromagnetic films [14]. The reflection of transversal spin electrons is approximately 50% [15]. The transmitted and reflected currents are polarized.

There are several approaches to describing spin transport. Spintronic devices are modeled using diffusion equations [16,17]. Multilayer ferromagnetic structures are modeled using quantum mechanical methods [18,19]. These models describe spin-transfer dynamics. But they cannot calculate chains. These models simulate the ferromagnetic state and transmit it as a set of spin current boundary conditions. More advanced models solve the joint system of equations for the ferromagnetic state and spin current [20]. Spin transport contributes to the demagnetization dynamics in metal ferromagnets [21,22]. This phenomenon is modeled using the Boltzmann equation for electron gas [23].

The Monte Carlo method is actively used to model spin transport [24]. The advantage of this approach is the ability to take into account the properties of electronic transport. This approach does not require additional assumptions that are used in the diffusion model [25]. A theoretical study of spin current in an antiferromagnetic film using the Heisenberg model and the Boltzmann equation showed the presence of a wide maximum near the Neel temperature [26].

The purpose of this article is to model the spin-transfer torque using the Monte Carlo method in three-layer structures.

## 2. Model and Computer Simulation

Our model studies a multilayer ferromagnetic film. The system consists of two parallel thin ferromagnetic films made of different materials. The magnetic properties of the materials are different. The first film has a thickness $D_{1}$. The material of this film is magnetically hard. The second film has a thickness $D_{2}$. The material of this film is magnetically soft. For thin films, the presence of easy axis magnetization has been experimentally proven [27]. The anisotropy axis is directed perpendicular to the film plane. A layer of nonmagnetic conductive material separates these films. The thickness of the interlayer is $D_{02}$. In addition, two nonmagnetic conductive films are present as coatings of ferromagnetic films. The thicknesses of these films are $D_{01}$ and $D_{03}$ (Figure 1). These films simulate electrical contacts. All films are parallel to the OXY plane in computer simulations.

The Ising model describes the magnetic properties of thin ferromagnetic films [28]. Each atom has a spin parallel to the OZ axis. The projection of spin S onto the OZ axis can take two values (1/2 or −1/2). The film material has a simple cubic lattice in this model. The crystal lattice period is a. The Hamiltonian of such a system is the sum of the paired interactions of spins.
(1)$$H_{0}=-J_{1}\sum_{\begin{array}{c} D_{01}\leq z_{1}<D_{01}+D_{1}\\ D_{01}\leq z_{2}<D_{01}+D_{1} \end{array}}S\left(x_{1},y_{1},z_{1}\right)S\left(x_{2},y_{2},z_{2}\right)-\;-J_{2}\sum_{\begin{array}{c} D_{01}+D_{1}+D_{01}\leq z_{1}<D_{01}+D_{1}+D_{01}+D_{2}\\ D_{01}+D_{1}+D_{01}\leq z_{2}<D_{01}+D_{1}+D_{01}+D_{2} \end{array}}S\left(x_{1},y_{1},z_{1}\right)S\left(x_{2},y_{2},z_{2}\right)$$

$S\left(x,y,z\right)$ is the spin of an atom at coordinates $\left(x,y,z\right)$. Atoms are located in nodes of the cubic lattice with period a. The first term in Hamiltonian describes the exchange interaction in the first film (area FM_1_ in Figure 1). $J_{1}$ is an exchange interaction constant for the material of the first film (magnetically hard ferromagnet FM_1_). The second term describes the exchange interaction in the second film (area FM_2_ in Figure 1). $J_{2}$ is an exchange interaction constant for the material of the second film (magnetically soft ferromagnet FM_2_). The material of the first film is magnetically hard and the material of the second film is magnetically soft, so the inequality is satisfied and $J_{1}>J_{2}$. The ferromagnetic films are separated by the nonmagnetic film (area NM_2_ in Figure 1). We consider the exchange interaction between films to be small and do not take it into account in calculations. In nonmagnetic films, atoms have zero spin (areas NM_1_, NM_2_, and NM_3_ in Figure 1).

The exchange interaction rapidly decreases with distance. Only pairs of nearest neighbors are counted in Hamiltonian (1). The crystal lattice period is a convenient unit of length in computer simulations. The model uses the value a=1. The coordinates of the pairs of interacting spins satisfy the conditions (2).
(2)$$\left(x_{i}=x_{j}\pm 1,\;y_{i}=y_{j},\;z_{i}=z_{j}\right)\;or\;\left(x_{i}=x_{j},\;y_{i}=y_{j}\pm 1,\;z_{i}=z_{j}\right)\;\;\;\;\;\;\;\;\;\;\;\;\;\;or\;\left(x_{i}=x_{j},\;y_{i}=y_{j}\pm 1,\;z_{i}=z_{j}\right).$$

Periodic boundary conditions are applied along the OX and OY axes. The film has dimensions L×L in the OXY plane. For spins with a coordinate $x_{i}=0$, neighboring to the left is a spin with a coordinate $x_{j}=L-1$ and vice versa. For spins with a coordinate $y_{i}=0$, neighboring below is a spin with a coordinate $y_{j}=L-1$ and vice versa. Finite-dimensional scaling theory calculates the properties of infinite systems from a set of finite systems [29]. This theory considers systems of different finite sizes L and approximates their properties to infinite systems.

The model is more convenient for computer simulation in relative values.
(3)$$R=\frac{J_{2}}{J_{1}},\;\;\;\;\;H_{1}=\frac{H_{0}}{J_{1}}$$

We write the Hamiltonian of the system in relative units.
(4)$$H_{1}=-\sum_{\begin{array}{c} D_{01}\leq z_{1}<D_{01}+D_{1}\\ D_{01}\leq z_{2}<D_{01}+D_{1} \end{array}}S\left(x_{1},y_{1},z_{1}\right)S\left(x_{2},y_{2},z_{2}\right)-\;-R\sum_{\begin{array}{c} D_{01}+D_{1}+D_{01}\leq z_{1}<D_{01}+D_{1}+D_{01}+D_{2}\\ D_{01}+D_{1}+D_{01}\leq z_{2}<D_{01}+D_{1}+D_{01}+D_{2} \end{array}}S\left(x_{1},y_{1},z_{1}\right)S\left(x_{2},y_{2},z_{2}\right).$$

The ratio of exchange integrals R is not greater than 1 (R ≤ 1).

The films have different phase transition temperatures $T_{1}$ and $T_{2}$. The phase transition temperature is directly proportional to the exchange interaction constant. The phase transition temperature in the first film is higher than in the second film ($T_{1}>T_{2}$), because $J_{1}>J_{2}$.

Magnetization in the ferromagnetic films is equal to the average value of spin in film. Magnetizations in the first ($m_{1}$) and second ($m_{2}$) films are their order parameter.
(5)$$m_{1}=\left(\sum_{0\leq z_{1}<D_{1}-1}S\left(x,y,z\right)\right)⧸\left(L^{2}D_{1}\right),m_{2}=\left(\sum_{D_{1}<z_{2}<D_{1}+D_{2}}S\left(x,y,z\right)\right)⧸\left(L^{2}D_{2}\right).$$

The behavior of the system at different temperatures is simulated using the Metropolis algorithm [29]. The relative temperature T is more convenient in computer simulations instead of the thermodynamic temperature t.
(6)$$T=\frac{k_{B}t}{J_{1}}$$

$k_{B}$ is the Boltzmann constant.

The magnetization of the films depends on the temperature. Fluctuations in magnetization are observed in ferromagnetic materials at a fixed temperature. Calculation of thermodynamic parameters for the system is performed by means of magnetization averaging over a large number of spin configurations. Averaged values of values are marked with angle brackets ($\langle \ldots \rangle$).

Finite-dimensional scaling theory calculates phase transition temperatures using fourth-order Binder cumulants [30].
(7)$$U_{1}=1-\frac{\langle m_{1}^{4}\rangle }{3{\langle m_{1}^{2}\rangle }^{2}},\;\;\;\;\;U_{2}=1-\frac{\langle m_{2}^{4}\rangle }{3{\langle m_{2}^{2}\rangle }^{2}}.$$

The Binder cumulant temperature plots for systems with different linear size L intersect at one point. The intersection point corresponds to the phase transition temperature. Angle brackets mean averaging over thermodynamic spin configurations.

We use a semiclassical description to move electrons. Electrons form an ideal electron gas. Each electron moves chaotically in the absence of an external electric field. Exposure to an electric field creates directional electron drift. The collective electron drift rate is proportional to the strength of the electric field E. Electrons interact with ferromagnetic film atoms by exchange interaction. The Monte Carlo method simulates the flow of electric current through a multilayer system [7]. The number of electrons $n_{e}$ in the model is half the number of atoms.
(8)$$n_{e}=\frac{1}{2}L^{2}\left(D_{1}+D_{2}+D_{01}+D_{02}+D_{03}\right).$$

This concentration corresponds to the average estimate for metals [31]. An electron at (x,y, z) has a spin σ(x,y,z). Projections of the spin onto the OZ axis are of interest in this model. An electron spin projection can have one of two values (1/2 or −1/2). The interaction between electron spins and atom spins is exchangeable. The Hamiltonian of this interaction equals the sum of the paired exchange interactions of spins.
(9)$$H_{0ea}=-\sum_{\begin{array}{c} \left(x,y,z\right)\\ \left(x_{1},y_{1},z_{1}\right) \end{array}}J_{ea}\left(x,y,z,x_{1},y_{1},z_{1}\right)S\left(x,y,z\right)\sigma \left(x_{1},y_{1},z_{1}\right).$$

Summation is performed over all electron–atom pairs. $J_{ea}$ is the exchange integral of the interaction between the spins of an electron and an atom.

Electrons move between the nodes of the crystal lattice. The distance between electrons and atoms is not a constant value. The exchange integral depends on the distance between the spins. We use a step function with exponential decrease with distance to approximate the exchange integral $J_{ea}$.
(10)$$J_{ea}\left(x,y,z,x_{1},y_{1},z_{1}\right)=J_{ea}(r)=\left\{\begin{array}{cc} J_{e}, & r\leq r_{0},\\ J_{e}exp\left(-\alpha \left(r-r_{0}\right)\right) & r>r_{0}. \end{array}\right.\;\;\;\;\;\;\;\;\;\;\;\;\;\;r=\sqrt{{\left(x-x_{1}\right)}^{2}+{\left(y-y_{1}\right)}^{2}+{\left(z-z_{1}\right)}^{2}}.$$

$r_{0}$ is the radius of action for the exchange forces between atoms and electrons. Parameter α shows the rate of decline for the exchange integral with distance and depends on the material. $J_{e}$ is constant. We use relative units for interaction of electrons and atoms.
(11)$$R_{ea}\left(r\right)=\frac{J_{ea}\left(r\right)}{J_{1}},\;R_{e}=\frac{J_{e}}{J_{1}},H_{ea}=\frac{H_{0ea}}{J_{1}}.$$

The Hamiltonian for the spin interaction of electrons and atoms has the form
(12)$$H_{ea}=-\sum_{\begin{array}{c} \left(x,y,z\right)\\ \left(x_{1},y_{1},z_{1}\right) \end{array}}R_{ea}\left(r\right)S\left(x,y,z\right)\sigma \left(x_{1},y_{1},z_{1}\right).\;r=\sqrt{{\left(x-x_{1}\right)}^{2}+{\left(y-y_{1}\right)}^{2}+{\left(z-z_{1}\right)}^{2}}\;\;\;\;\;\;\;\;\;\;\;\;\;\;\;\;\;R_{ea}(r)=\left\{\begin{array}{cc} R_{e}, & r\leq r_{0},\\ R_{e}exp\left(-\alpha \left(r-r_{0}\right)\right) & r>r_{0}. \end{array}\right..$$

An electric field creates an electric current in the conductor. ${\vec{E}}_{0}$ is a uniform electric field. If an electron moves to a vector $∆\vec{r}$, then its energy in the electric field changes by $∆H_{0eE}$.
(13)$$∆H_{0eE}=-e{\vec{E}}_{0}∆\vec{r}.$$

e is the charge of an electron. We use the relative value of the electric field strength.
(14)$$\vec{E}=e{\vec{E}}_{0}/J_{1}.$$

The relative change in energy in the external electric field is
(15)$$∆H_{eE}=∆H_{0eE}/J_{1}=-\vec{E}∆\vec{r}.$$

The total change in the energy of the electron when moving to the vector $∆\vec{r}$ is equal to
(16)$$∆H_{e}=∆H_{eE}+∆H_{ea}=-\vec{E}∆\vec{r}-\;-\sum_{\begin{array}{c} \left(x,y,z\right)\\ \left(x_{1},y_{1},z_{1}\right) \end{array}}R_{ea}\left(r\right)S\left(x,y,z\right)\sigma \left({\vec{r}}_{1}\right)+\sum_{\begin{array}{c} \left(x,y,z\right)\\ \left(x_{1},y_{1},z_{1}\right) \end{array}}R_{ea}\left(r\right)S\left(x,y,z\right)\sigma \left({\vec{r}}_{1}+∆\vec{r}\right),\;\;\;\;\;\;\;\;\;\;\;\;\;\;{\vec{r}}_{1}=\left(x_{1},y_{1},z_{1}\right).$$

The energy of interaction between the spins of electrons and atoms must also be considered in Hamiltonian (4). We record the final form of the Hamiltonian, taking into account the interaction with electrons. Interaction with polarized electric current affects the ordering of atomic spins.
(17)$$H=-\sum_{\begin{array}{c} 0\leq z_{1}<D_{1}-1\\ 0\leq z_{2}<D_{1}-1 \end{array}}S\left(x_{1},y_{1},z_{1}\right)S\left(x_{2},y_{2},z_{2}\right)-\;-R\sum_{\begin{array}{c} D_{1}<z_{1}<D_{1}+D_{2}\\ D_{1}<z_{2}<D_{1}+D_{2} \end{array}}S\left(x_{1},y_{1},z_{1}\right)S\left(x_{2},y_{2},z_{2}\right)-\;-\sum_{\begin{array}{c} \left(x,y,z\right)\\ \left(x_{1},y_{1},z_{1}\right) \end{array}}R_{ea}\left(r\right)S\left(x,y,z\right)\sigma \left(x_{1},y_{1},z_{1}\right)$$

This model does not take into account the energy of the exchange interaction between electrons, because it is much less than the energy of interaction with atoms.

Our model also considers the surface potential at the conductor boundary. The surface potential inhibits the free drift of electrons between metals. The lack of surface potential leads to the redistribution of electrons between metals in the absence of electric current. We use a rectangular potential barrier to model surface potential. The barrier width is 0.5. The height of the potential barrier is $U_{0}$. Barrier height is measured in units of $J_{1}$.

We use the Monte Carlo method to describe the motion of electrons in the crystal lattice [7]. One step of the algorithm consists of trying to shift the electron to a random vector $∆\vec{r}$. The change in electron energy is $∆H_{e}$, calculated for this step by the formula (16). If the energy change is less than 0 ($∆H_{e}<0$), then a new electron position is adopted. If the energy change is greater than 0 ($∆H_{e}>0$), then the new position of the electron is taken with probability $P_{e}$.
(18)$$P_{e}=exp\left(-∆H_{e}/T\right).$$

One iteration of the algorithm involves trying to shift each electron. The electron has zero velocity after each movement. This approximation simulates energy loss from colliding with atoms.

Electrons are randomly distributed over the crystal lattice in the initial state. Electron spins also have a random direction with equal probability. If the electron goes beyond the boundaries of the system, then it is transferred to the opposite side.

The computer simulation consists of repeated cycles. The first part of the cycle performs $N_{1}$ iterations to find the equilibrium state of the spins of atoms. The second part of the cycle performs $N_{2}$ iterations to move electrons along the crystal lattice. The total number of cycles is $N_{3}$. This approach simulates the passage of an electric current, taking into account the change in the configuration of the spins of atoms.

## 3. Results

We investigate spin transport in films with thickness $D_{1}=8\;ML$, $D_{2}=8\;ML$, $D_{01}=8\;ML$, $D_{02}=8\;ML$, and $D_{03}=8\;ML$. The number of monoatomic layers (ML) is the unit of measurement for film thickness. The ratio of exchange interaction constants is R=0.5 in a computer experiment. This value creates a sufficiently large temperature interval between the Curie points for the two films. The first stage of computer simulation examines the phase transition temperature in both films. Finite-dimensional scaling theory requires the study of systems with different linear sizes. We simulate systems with sizes from L=24 ML to L=64 ML in ∆L=8 ML increments. Calculations give the ferromagnetic phase transition temperatures in films as $T_{1}=4.2$ and $T_{2}=2.2$. Further modeling is performed at temperature $T<T_{2}$. Both films are in ferromagnetic phase at this temperature. The second film is most sensitive to external influences at temperatures close to $T_{2}$. The computer experiment is performed at a temperature T=1.9.

Spin transport studies are performed on L=64 ML films. The model uses the parameters $R_{e}=0.3$, $r_{0}=0.5$, and a=1.5 for the exchange interaction between electrons and atoms. Computer simulation shows that changing these parameters does not qualitatively change the behavior of the system. The electric field varies from E=0 to E=0.2 in ∆E=0.01 increments. The number of Monte Carlo steps is $N_{1}=N_{2}=10^{5}$. The number of iterations is $N_{3}=10^{3}$. The potential barrier at the film boundary is $U_{0}=0.5$.

The electric field creates electron motion in the positive direction of the OZ axis in a computer simulation. Electrons transfer spin from a magnetically hard film to a magnetically soft film. The spins of the magnetically hard film are oriented in the positive direction of the OZ axis in the absence of an electric field. Magnetization in this film has a positive value. The spins in the second film are oriented in the negative direction of the OZ axis in the absence of an electric field. The magnetization in the second film is negative. This magnetization orientation is maintained at temperatures below the Curie point in the absence of electrical current. The electric current passes from the nonmagnetic film to the magnetically hard film and is polarized. Spin-up electrons interact with the spins of atoms in a magnetically soft film. If there are many polarized electrons, then there is a magnetization reversal in the magnetically soft film. A computer experiment studies the dependence of magnetization in a magnetically soft film on the strength of an electric field. Figure 2 shows a plot of this relationship.

The magnetization plot for the magnetically soft film versus the electric field is stepwise. At low electric field values, a negative magnetization value is maintained in the second film. Magnetization switching in the magnetically soft film occurs under the influence of the electric current. Switching is implemented in the form of a change in the magnetization sign. There is a minimum value of the electric field strength $E_{c}$, creating sufficient electric current to change the magnetization sign.

The real experiment measures the dependence of magnetization on the current strength in the system. The number of electrons $n_{e}$ passing through the interface of the films determines the electric current density $j_{0}$ in this model. Electrons moving in the positive direction of the OZ axis increase the value of $n_{e}$. Electrons moving in the negative direction of the OZ axis decrease the value of $n_{e}$.
(19)$$j_{0}=\frac{en_{e}}{S∆t}.$$

S is the cross-sectional area of the conductor. S is equal to the area of the films in our system ($S=L^{2}$). ∆t is the current flow time. We use one iteration as a unit of time. The current flow time is equal to the number of iterations ($∆t=N_{3}$). The number of passed electrons *n_e_* is equal to the difference in the number of electrons crossing the film boundary in the positive and negative directions. We use the relative magnitude of the current density j for uniformity of the model.
(20)$$j=\frac{j_{0}}{eJ_{1}}=\frac{n_{e}}{L^{2}N_{3}}.$$

Figure 3 shows a plot of the current strength versus the electric field strength.

The dependence of the current on the electric field strength is nonlinear. This dependence is due to the effect of the electric current on the magnetization in the second film. Resistance to electric current depends on the exchange interaction between the spins of atoms and electrons. This interaction changes because magnetization changes. Resistance depends not only on the strength of the electric field but also on the strength of the current. This factor results in a nonlinear voltammeter characteristic. Figure 4 is a plot of the magnetization in the second film versus the current in the system.

The dependence of magnetization $m_{2}$ on current j is stepwise. Current $j_{c}$ completely demagnetizes magnetically soft film. If the current exceeds the value $j_{c}$, the magnetization of the free film is inverted.

The demagnetization and inversion of the magnetization in the free film is due to the polarization of the electron gas. The polarization of electrons P in films depends on the number of electrons with spin-up or -down.
(21)$$P=\frac{n_{↑}-n_{↓}}{n_{↑}+n_{↓}}.$$

$n_{↑}$ is the number of spin-up electrons. $n_{↓}$ is the number of spin-down electrons. The exchange interaction between electrons and atoms leads to a change in the polarization of the electron gas. Figure 5 shows plots of the polarization for the electron gas in the first film $P_{1}$ and in the second film $P_{2}$ versus the current.

The electron gas in the first film becomes completely polarized with a low electric current. The polarization of the electron gas in the second film becomes nonzero at a sufficiently high current. Demagnetization of the free plate occurs at current *j_c_*. The polarization of the electron gas in this case exceeds 0.7.

It should be noted that polarization of electron gas occurs not only in ferromagnetic films but also in nonmagnetic films. Figure 6 shows a plot of the polarization in the film through which the electric current is applied to the magnetically hard ferromagnetic film.

The polarization of the electron gas in the current-supplying film is opposite to polarization in the first ferromagnetic film. This indicates surface electron filtration at the film interface.

## 4. Conclusions

Our model describes the change in magnetization in a ferromagnetic film using polarized electric current with the Monte Carlo method. Polarization of the electric current occurs at the boundary of the conductive nonmagnetic film and the fixed ferromagnetic film. The polarized electron gas moves under the influence of an electric field and transfers the spin into a magnetically soft film. The interaction between the electron spins and the atom spins of the second film leads first to the demagnetization of this film and then to a change in the magnetization sign. This phenomenon is observed experimentally [13]. In some experiments, spin transfer magnetizes a ferromagnetic film that had a zero magnetic moment initially. However, the model used cannot form a stable state with zero magnetization below the Curie point. Reproduction of simple film magnetization cannot be implemented in this model. This disadvantage is common to all spin models of magnetic materials.

We make a few comments about the choice of model parameters. The constant of the exchange interaction between electrons and atoms is quite large in our model. This selection is based on the limited dimensions of the system being modeled. Smaller values of the exchange constant require more current to influence the magnetic moment of the thin film. Attempts to create a large electric current are limited by the number of electrons that can be used. A large electric field strength does not create a large electric current due to a lack of electrons. Other model parameters can vary over a wide range. The qualitative picture of the phenomenon remains unchanged.

The existence of a minimum current for system magnetization is an experimental fact [13]. Our model defines this current in relative units. Conversion to absolute units allows you to determine the order of the magnitude of this current ($j_{c}\sim 10^{7}A/{cm}^{2}$). This value in order of magnitude is close to the experimental value [13].

The model in this article considers only the general principles of spin-transfer torque simulation. Consideration of specific materials is possible by complicating the crystal lattice and refining the model parameters.

In our model, the motion of electrons is free. The interaction of mobile spin carriers occurs only with ferromagnetic atoms. However, real materials contain structural defects that act as traps for mobile particles and reduce their mobility [32,33]. This effect can result in additional electrical resistance. Structural defects also affect the magnetization of the system and change the Curie temperature, which also affects the interaction between the spins of electrons and atoms. Defects on the film interface should have the greatest impact. Addressing these issues requires substantial model development and extensive research.

Our model describes spin-transfer torque only in ferromagnetic metals. Other materials (antiferromagnets or semiconductors) require model processing. The main assumption of the model is the free movement of electrons. Any restrictions on the mobility of spin-carrying particles require refinement of the model.

## Figures and Tables

**Figure 1 materials-16-06728-f001:**
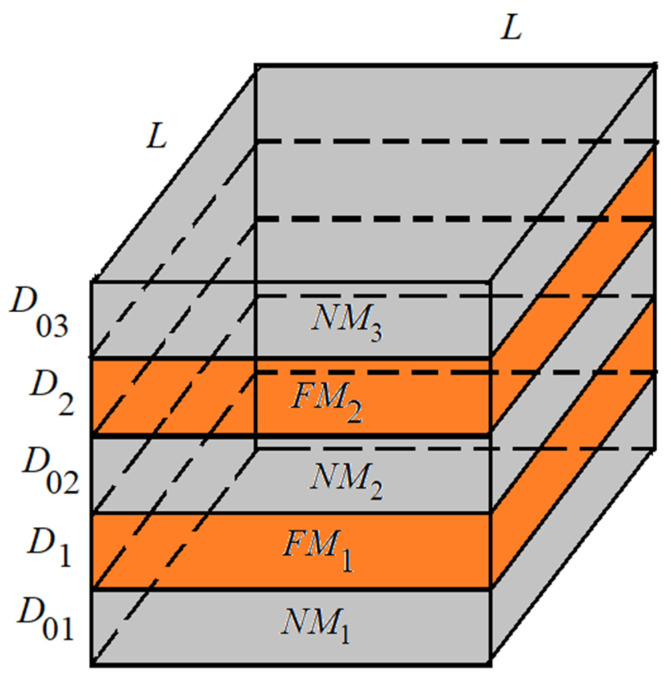
System geometry. NM is a nonmagnetic material. ${FM}_{1}$ is a magnetically hard material. ${FM}_{2}$ is a magnetically soft material.

**Figure 2 materials-16-06728-f002:**
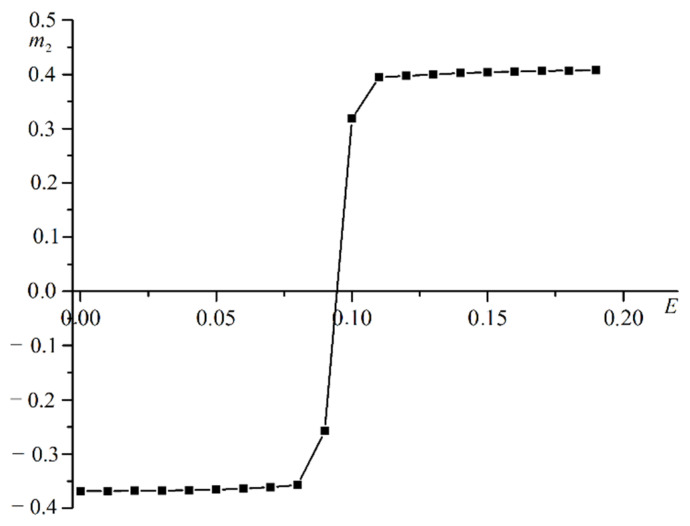
Plot of magnetization in magnetically soft film $m_{2}$ versus electric field strength E.

**Figure 3 materials-16-06728-f003:**
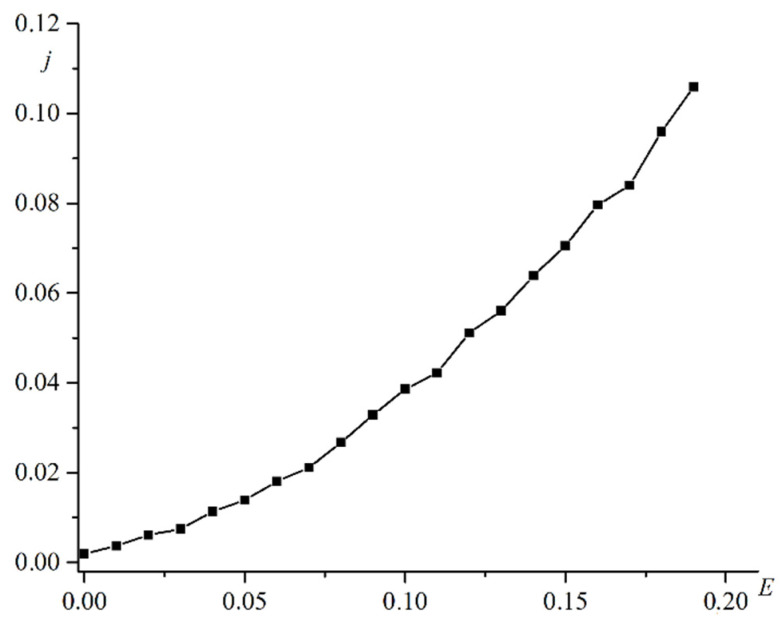
Plot of electric current density j versus electric field strength E.

**Figure 4 materials-16-06728-f004:**
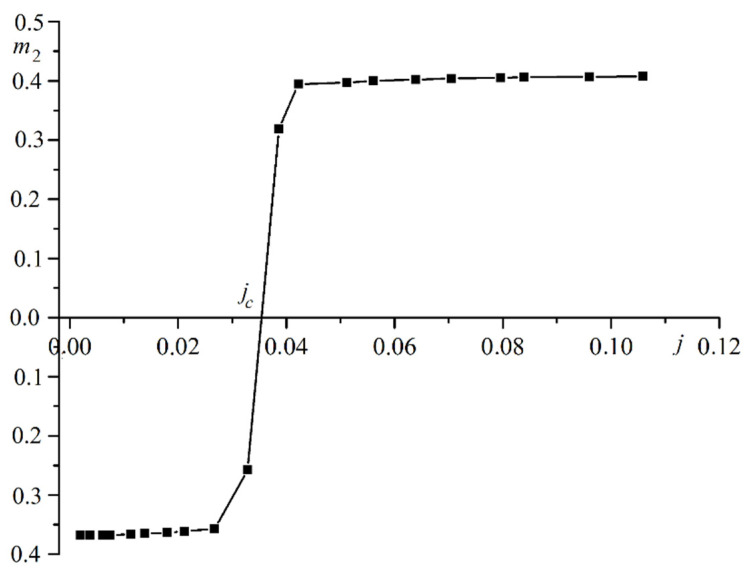
Plot of the magnetization in the second film $m_{2}$ versus the current in the system j.

**Figure 5 materials-16-06728-f005:**
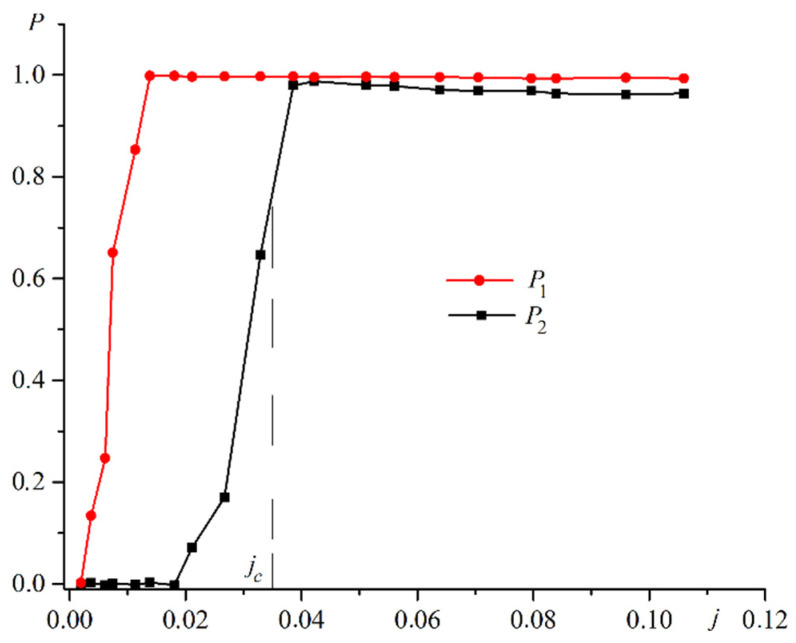
Plots of the polarization for the electron gas in the first film $P_{1}$ (red line) and in the second film $P_{2}$ (black line) against the current j.

**Figure 6 materials-16-06728-f006:**
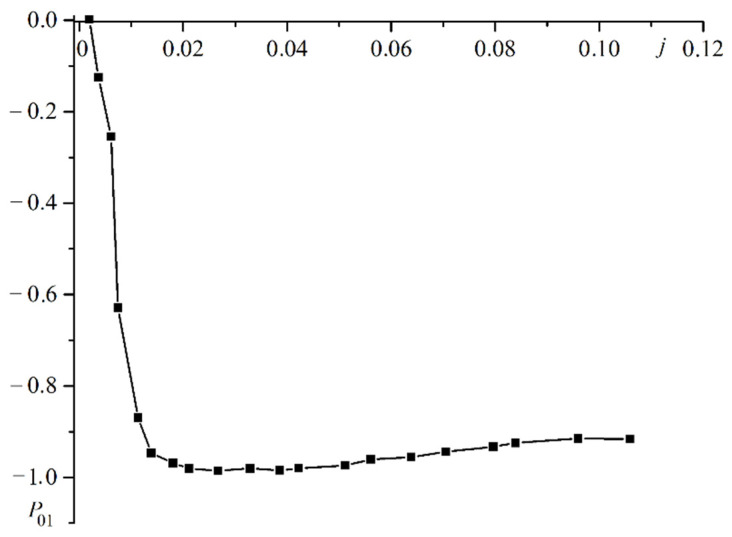
Plot of the polarization of the electron gas $P_{01}$ in the film supplying the electric current versus the current j.

## Data Availability

Not applicable.

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
