# Peer review of "Monte Carlo Computer Simulations of Spin-Transfer Torque"

_materials, 2023, doi:10.3390/ma16206728_

Round 1
Reviewer 1 Report
I recommend publication after significant revision. Details are as follows:
1
The authors used Monte-Carlo simulation in this study; However, this is not reflected in the title of this article.
2
Lines 52-54 describe the physics behind spin transfer. I think you should include a graphical schematic.
3
The dimension of Figure 2 is hard to follow with the description in lines 86-99.
4
Would you like to explain why 'D1-1' in the summation? How is it referred to as Figure 2?
Equations (1) and (2) may need more discussion.
5
From Equations (5), I expect that m1 and m2 are constants as the averages of S over the sample. However, Equations (7) include <m1> and <m2>. What do they refer to? Usually, we assume that <> means the mean values.
6
The Monte Carlo simulation was applied for this study. Can you briefly discuss how it is applied to the spin transfer? Formulas (integral, probability distribution, etc.) Algorithms, implementation by code or software or open-source? How much time is needed? How heavy is the simulation?
7
In Equation (10), \alpha is not defined.
8
Equations (13) and (16) have two dots at the end. Would you like to double-check?
9
The description in lines 177-184 may require a graphical schematic.
10
In the result (lines 186-197), temperatures T1 and T2 have no unit. Are they normalized?
11
The description in lines 205-217 may require a graphical schematic.
12
All figures do not include units. I think you may want to verify.
The authors may need to rewrite the article.
Author Response
Dear reviewer!
We thank you for your constructive comments. We provide answers to your questions and comments.
- The authors used Monte-Carlo simulation in this study; However, this is not reflected in the title of this article.
Answer: We corrected the title.
“Spin-Transfer Torque: Monte-Carlo computer simulations”
- Lines 52-54 describe the physics behind spin transfer. I think you should include a graphical schematic.
Answer: These lines are given in the introduction to the article. We provide links to publications describing this phenomenon. The additional figure will weigh down the introduction.
- The dimension of Figure 2 is hard to follow with the description in lines 86-99.
Answer: We have supplemented the description in these lines with references to Figure 1.
“Atoms are located in nodes of cubic lattice with period . The first term in Hamiltonian describes the exchange interaction in the first film (area FM1 in Figure 1). is an exchange interaction constant for the material of the first film (magnetically hard ferromagnet FM1). The second term describes the exchange interaction in the second film (area FM2 in Figure 1). is an exchange interaction constant for the material of the second film (magnetically soft ferromagnet FM2). The material of the first film is magnetically hard and the material of the second film is magnetically soft, so the inequality is satisfied . Ferromagnetic films are separated by non-magnetic film (area NM2 in Figure 1). We consider the exchange interaction between films small and do not take it into account in calculations. In non-magnetic films, atoms have zero spin (areas NM1, NM2 and NM3 in Figure 1).”
4.Would you like to explain why 'D1-1' in the summation? How is it referred to as Figure 2?
Equations (1) and (2) may need more discussion.
Answer: We corrected the summation limits in formulas (1) and (4).
In the first term, the expression for the summation limits is D_01≤z_1<D_01+D_1, in the second term, the summation limits are D_01+D_1+D_01≤z_1<D_01+D_1+D_01+D_2.
Equations (1) and (2) define the standard Hamiltonian of the Ising model. It's well known. Of interest are only the parameters for this Hamiltonian. We consider the description of the parameters sufficient to reproduce our results.
5.From Equations (5), I expect that m1 and m2 are constants as the averages of S over the sample. However, Equations (7) include <m1> and <m2>. What do they refer to? Usually, we assume that <> means the mean values.
Answer: We added the phrase:
“The magnetization of the films depends on the temperature. Fluctuations in magnetization are observed in ferromagnetic materials at a fixed temperature. Calculation of thermodynamic parameters for the system is performed by means of magnetization averaging over a large number of spin configurations. Averaged values of values are marked with angle brackets ().”
- The Monte Carlo simulation was applied for this study. Can you briefly discuss how it is applied to the spin transfer? Formulas (integral, probability distribution, etc.) Algorithms, implementation by code or software or open-source? How much time is needed? How heavy is the simulation?
Answer: The Monte Carlo algorithm is described at the end of Section 2. We indicated at the beginning of Section 3 the modeling parameters and the number of Monte Carlo steps. There is no standard software for solving such problems. We have independently implemented the settlement software.
- In Equation (10), \alpha is not defined.
Answer: We added the phrase:
“Parameter shows the rate of decline for the exchange integral with distance and depends on the material.”
- Equations (13) and (16) have two dots at the end. Would you like to double-check?
Answer: All formulas were rechecked. Thank you.
- The description in lines 177-184 may require a graphical schematic.
Answer: The description in lines 177-184 gives details for the software implementation of the algorithm. These phrases are enough for specialists when reproducing results. A graphical diagram is not required here.
- In the result (lines 186-197), temperatures T1 and T2 have no unit. Are they normalized?
Answer: Temperatures are normalized by formula (6).
- The description in lines 205-217 may require a graphical schematic.
Answer: We believe that the graphical schematic to the process described in lines 205-207 will be too simple. The description is step-by-step. Each step of the process is quite simple. The graphic schematic will consist of a large number of independent details and may introduce confusion into the article.
- All figures do not include units. I think you may want to verify.
Answer: Computer simulations use relative units. The definition of relative units through absolute ones is given in formulas (3), (6), (11) and (15) when describing the model.
Reviewer 2 Report
I have read the submitted article entitled “Spin-Transfer Torque: Computer Simulations” by S. V. Belim et al submitted to Materials/MDPI. A variety of emerging storage solutions have been researched to either improve or substitute spin-transfer torque devices. The authors have discussed the magneto-resistance effect, where the magnetic field is used to switch the magnetization of a magnetic film. Computer simulations have been carried out to model the spin-transfer torque by the Monte Carlo method in three layers such as free, spacer, and pinning. The paper is written well, and my specific points that require some attention are given below.
· The basic structure of a spin-valve with clear illustrations labeling Free space must be illustrated.
· Provide the basic explanation of giant magnetoresistance with the difference in the scattering process of spin-polarised electrons.
· How to reduce the overall consumption of the device?
· The program that has been chosen to simulate the devices must be elaborated while providing the parameter values.
· The appropriate units must be provided in the X/Y-axes for all the figures.
· Although the resistance to electric current depends on the exchange interaction between the spins of atoms and electrons the presence of traps/defects can cause disruptions in the flow (leading to more significant changes in the interaction), Justify this with reference to the literature discussed in /doi.org/10.3390/ma16134691; and doi.org/10.3390/nano12244414).
· The limitations of the current model must be stated.
· The section conclusion must be precise rather than story-telling.
Author Response
Dear reviewer!
We thank you for your constructive comments. We provide answers to your questions and comments.
- The basic structure of a spin-valve with clear illustrations labeling Free space must be illustrated.
Answer: The model in question describes more than a spin valve. We model a phenomenon used in multiple spintronic devices. The narrowing of the subject matter to the spin valve is not quite consistent with the purpose of the simulation. We believe that such an addition will distort the setting of the task in the article. The overall design of the spin valve is well known. Description of the design of this device may reduce the interest of those skilled in the art.
- Provide the basic explanation of giant magnetoresistance with the difference in the scattering process of spin-polarised electrons.
Answer: The effect of giant magnetoresistance is not the main aim of this article. We do not investigate the dependence of electrical resistance on film magnetization. A detailed description of this effect may mislead the reader about the task being solved in the article. Monte Carlo modeling of giant magnetoresistance was implemented earlier in the articles [24,25].
- How to reduce the overall consumption of the device?
Answer: The model does not allow calculating the energy effect at this stage. It is necessary to refine the model taking into account not only direct current, but also reverse current. This task is in our plans in the near future and is the topic of a separate article.
- The program that has been chosen to simulate the devices must be elaborated while providing the parameter values.
Answer: In the program, all parameters are set in relative units. The transition to specific materials is possible with known values ​ ​ of exchange integrals. Exchange integrals are determined either experimentally or on the basis of quantum mechanical calculations from the first principles.
- The appropriate units must be provided in the X/Y-axes for all the figures.
Answer: In a computer model, all quantities are relative and have no dimension. The definition of relative units through absolute ones is given in formulas (3), (6), (11) and (15) when describing the model.
- Although the resistance to electric current depends on the exchange interaction between the spins of atoms and electrons the presence of traps/defects can cause disruptions in the flow (leading to more significant changes in the interaction), Justify this with reference to the literature discussed in /doi.org/10.3390/ma16134691; and doi.org/10.3390/nano12244414).
Answer: We added the phrase:
“In our model, the motion of electrons is free. The interaction of mobile spin carriers occurs only with ferromagnetic atoms. However, real materials contain structural defects that act as traps for mobile particles and reduce their mobility [33,34]. This effect can result in additional electrical resistance. Structure defects also affect the magnetization of the system and change the Curie temperature, which also affects the interaction between the spins of electrons and atoms. Defects on the film interface should have the greatest impact. Addressing these issues requires substantial model development and extensive research.”
- The limitations of the current model must be stated.
Answer: We added the phrase:
“Our model describes Spin-Transfer Torque only in ferromagnetic metals. Other materials (antiferromagnets or semiconductors) require model processing. The main assumption of the model is the free movement of electrons. Any restrictions on the mobility of spin-carrying particles require refinement of the model.”
- The section conclusion must be precise rather than story-telling.
Answer: Dear reviewer, there are various approaches to writing scientific articles. The author independently chooses the style of presentation. There is no strict algorithm for writing scientific articles. The conclusion exists in order to summarize the studies performed in the article and discuss the possible development of research. The preferences of readers also vary. We considered that for this article a simple statement of the results in the conclusion would not be interesting for readers.
Round 2
Reviewer 1 Report
The authors have clarified most of my questions and I recommended publication.
The English is acceptable.